# Kidney Transplantation in Patients with Multiple Myeloma: Current Evidence, Challenges, and Future Directions

**DOI:** 10.3390/ijms26199358

**Published:** 2025-09-25

**Authors:** Natacha Rodrigues, Manuel Silva, Carolina Branco, Sofia Barreto, Telma Pais, José António Lopes

**Affiliations:** 1Department of Nephrology and Renal Transplantation, Unidade de Saúde Local Santa Maria, 1649-028 Lisbon, Portugal; manueljoaosantoslopessilva@gmail.com (M.S.); carolina.branco@chln.min-saude.pt (C.B.); telmapais.m@gmail.com (T.P.); 2Faculty of Medicine, University of Lisbon, 1649-028 Lisbon, Portugal; sofiacbarretov@gmail.com

**Keywords:** multiple myeloma, end stage renal disease, kidney transplantation

## Abstract

Renal involvement is an important complication of multiple myeloma (MM) and is related not only to worse clinical outcomes but also to lower quality of life, particularly when progressing to end-stage renal disease. Traditionally, MM patients were not considered eligible for kidney transplant; however, these paradigms are changing. The new era of MM therapies brought proteasome inhibitors, immunomodulatory drugs, monoclonal antibodies, and, most recently, cellular therapies, leading to longer survival and sustained hematological responses. Knowledge of cytogenetic abnormalities has helped risk stratification. These advances result in the identification of patients who achieve durable remission and may benefit from kidney transplant programs as an option for renal replacement therapy. Reported 5-year allograft survival ranges from 50 to 66%, progression-free survival is 44%, and overall survival is 61%, depending on pre-transplant remission depth. This review summarizes updated available evidence regarding kidney transplants in MM, proposes evidence-based eligibility criteria for kidney transplantation in this population, and outlines therapeutic strategies for long-term follow-up. In conclusion, kidney transplantation may be a feasible option for carefully selected MM patients achieving deep and sustained remission, though prospective data are still needed.

## 1. Introduction

Kidney transplantation represents the best option for renal replacement therapy in eligible patients with end-stage renal disease (ESRD), as it is associated with improved survival and quality of life compared with dialysis [1]. Traditionally, hematological malignancies, including multiple myeloma (MM), were considered contraindications because of the increased risk of relapse and post-transplant complications under long-term immunosuppression [2,3,4]. However, MM is both the most frequent non-urologic malignancy associated with kidney injury and a major cause of ESRD [5,6,7]. Recent advances in therapy—including proteasome inhibitors, immunomodulatory drugs, monoclonal antibodies, cellular therapies, and hematopoietic stem cell transplantation (HSCT)—have markedly improved survival. Also, with the current knowledge on cytogenetics and high-risk profiles, it is now possible to stratify patients according to their risk of relapse and disease progression. As a result, carefully selected MM patients who achieve sustained remission may now be considered for kidney transplantation, prompting the need to revisit eligibility criteria for this population [8].

We present a comprehensive review of the literature on this subject and summarize the available evidence regarding kidney transplantation in MM. We propose evidence-based eligibility criteria for kidney transplantation in this population and outline therapeutic strategies for long-term follow-up. By doing so, we hope to ultimately allow carefully selected patients with MM to benefit from the advantages of kidney transplantation.

## 2. Methods

We performed a narrative review of the literature focusing on ESRD and kidney transplantation in patients with MM and related plasma cell dyscrasias. A systematic search was conducted in PubMed and Embase, identifying publications in English, Portuguese, or Spanish from January 1975 through June 2025. We used combinations of the following keywords: “multiple myeloma,” “plasma cell dyscrasia,” “renal transplantation,” “kidney transplantation,” “end-stage renal disease,” and “monoclonal gammopathy”. We included case reports, case series, cohort studies, registry analyses, and review articles that provided clinical outcomes or guidance on patient selection, timing, or management of kidney transplantation in this population. Given the heterogeneity and limited number of available studies, we synthesized the evidence in a narrative format, supported by summary tables.

## 3. Renal Involvement in Multiple Myeloma

Renal involvement is a common and clinically significant feature of MM. It may be present at diagnosis or develop during the course of disease and arises from several distinct pathophysiological mechanisms [9].

### 3.1. Pathophysiological Mechanisms

#### 3.1.1. Light Chain Cast Nephropathy

Light chain cast nephropathy (LCCN), a myeloma-defining event, results from the overproduction of monoclonal free light chains (FLCs) that overwhelm the absorptive capacity of proximal tubular cells [10]. Within the proximal tubule, FLCs trigger oxidative stress, apoptosis, and pro-inflammatory/pro-fibrotic signaling, leading to tubular injury and interstitial fibrosis.

In the loop of Henle and distal tubule, unabsorbed FLCs interact with uromodulin (Tamm–Horsfall protein), forming obstructive casts that induce inflammation and tubular atrophy [10,11,12]. The affinity of FLCs for uromodulin varies, depending on structural determinants within the variable domain, such as the complementarity-determining region 3 (CDR3) of the variable domain of light chains, and partly explains the heterogeneity of LCCN presentation [12,13].

Clinically, the risk of LCCN rises with serum FLC levels above ~1500 mg/L and higher urinary FLC excretion [14]. Additional risk factors include low urine flow (e.g., from non-steroidal anti-inflammatory drugs, loop diuretics) and increased urinary sodium or calcium concentrations, which promote cast formation [12].

#### 3.1.2. Monoclonal Immunoglobulin Deposition and Aggregation Disorders

Beyond LCCN, additional renal impairment may result from the deposition of monoclonal immunoglobulins—either intact molecules, fragments, or aggregated products. When myeloma-defining criteria are not fulfilled, these conditions fall under the spectrum of monoclonal gammopathy of renal significance (MGRS) [11].

**Amyloidosis (AL, AH, and AHL types).** Misfolded immunoglobulin light chains, heavy chains, or both may aggregate into amyloid fibrils, depositing in glomeruli, vessels, and interstitium. These deposits provoke local inflammation and disrupt tissue architecture, typically presenting with nephrotic syndrome and progressive renal failure [10,15].

**Monoclonal fibrillary glomerulonephritis.** In rare cases, IgG deposits with light-chain restriction form non-amyloid fibrils, inducing glomerulonephritis [15].

**Proliferative glomerulonephritis with monoclonal immunoglobulin deposits (PGNMID).** Usually caused by IgG, this entity predominantly affects glomeruli, as the large immunoglobulin molecules cannot reach tubules [12].

**Microtubular deposits.** Immunotactoid glomerulonephritis and cryoglobulinemic glomerulonephritis are characterized by organized microtubular deposits. The former typically presents with isolated renal disease, while the latter often has systemic manifestations such as skin, neurological, and joint involvement [12,15].

#### 3.1.3. Renal Involvement Without Monoclonal Immunoglobulin Deposits

Renal injury in MM may occur through complement dysregulation and microvascular injury. When myeloma-defining criteria are absent, such lesions are also considered part of the MGRS spectrum [10,15].

**C3 glomerulopathy.** This condition is associated with alternative pathway dysregulation and glomerular C3 deposition. The higher-than-expected frequency of monoclonal gammopathy in these patients suggests a pathogenic link, as treatment of the underlying clonal disorder can improve renal outcomes [16].

**Thrombotic microangiopathy.** Both acute and chronic lesions involving glomeruli, arterioles, and small arteries have been reported in MM, leading to progressive renal impairment [12].

#### 3.1.4. Indirect Mechanisms

Hypercalcemia is present in 30% of cases at diagnosis and is a common cause of acute kidney injury (AKI) in MM patients. Hypercalcemia can cause volume contraction and renal vasoconstriction, leading to pre-renal AKI. It is also associated with nephrocalcinosis, with calcium deposits in renal parenchyma and tubular damage [10,12].

Additional contributors to renal dysfunction in MM are infectious complications, tumor lysis syndrome use of potentially nephrotoxic drugs (e.g., non-steroidal anti-inflammatory drugs, aminoglycosides, and bisphosphonates) [12]. Comorbidities such as diabetes, atherosclerotic vascular disease, and heart failure are also important known risk factors [11].

### 3.2. Epidemiology of Kidney Injury in Multiple Myeloma

Renal impairment is highly prevalent in MM, with reported frequencies at diagnosis ranging from 20% to 50%, depending on the definition used [9,17,18]. In the Australia and New Zealand Myeloma Registry, one-third of patients presented with reduced estimated glomerular filtration rate (eGFR) at diagnosis [17]. Historical series report even higher rates, up to 50% [19]. Variability in prevalence largely reflects differences in diagnostic thresholds, such as serum creatinine > 2 mg/dL versus eGFR < 60 or <40 mL/min/1.73 m^2^. Recent studies also suggest that high-risk cytogenetics may predispose to AKI in MM [20].

### 3.3. Prognostic Implications

Renal impairment in MM is consistently associated with poorer outcomes. In a meta-analysis of Mohyuddin et al. (2021) including >21,000 patients, renal impairment was present in 28.8% and correlated with increased risk of progression and mortality across both newly diagnosed and relapsed/refractory settings [9]. Chen et al. (2020) concluded that severe renal impairment, defined as an eGFR < 30 mL/min/m^2^, was significantly associated with decreased overall survival in multivariable analysis [21].

Importantly, renal recovery after anti-MM therapy is a strong prognostic factor. Patients who fail to achieve renal response have markedly reduced survival compared with those who recover kidney function [21]. In a population-based registry, Courant et al. (2021) found that combined hematological and renal response was strongly associated with improved 1-year overall survival, highlighting the interdependence of hematological control and renal recovery [22].

## 4. Advances in Myeloma Therapy and Impact on Transplant Candidacy

In the last two decades, we have witnessed a revolution in the therapeutic approach to MM. The development of a variety of new drugs and techniques has translated into longer overall survival and sustained hematological responses, along with improved renal outcomes. This new reality has redefined MM as a chronic disease for many patients, and this is the basis for reconsidering kidney transplant as an option for carefully selected patients.

### 4.1. Evolution of Multiple Myeloma Therapy

From oral melphalan in combination with prednisone as first-line therapy for MM in the 1960s, followed by IMiD thalidomide addition, autologous HSCT in the 1980s, the introduction of proteasome inhibitors in the 2000s, and new generations of IMiD anti-CD38 monoclonal antibodies in the 2010s, evolution has been constant, resulting in continuous improved outcomes for MM patients.

**Proteasome inhibitors:** The first-generation FDA-approved proteasome inhibitor, bortezomib, revolutionized treatment by providing rapid cytoreduction. Metabolized by the CYP3A4 and CYP2C19 enzymes in the liver, it can be used in patients with severe renal impairment, including those on dialysis [11]. Second-generation proteasome inhibitors (carfilzomib and ixazomib) have been associated with additional efficacy and lower neurotoxicity, though concerns regarding cardiovascular toxicity have been raised [23].

**Immunomodulatory drugs:** Thalidomide, lenalidomide, and pomalidomide have significantly improved progression-free and overall survival in these patients. However, lenalidomide and pomalidomide have renal excretion and need careful dosing in chronic kidney disease patients [11]. Also, lenalidomide has a limited role after kidney transplant, as it has been associated with a higher risk of acute rejection (more references to this subject in Section 6).

**Monoclonal antibodies:** Monoclonal antibodies can be used in MM treatment to target surface markers on cancer cells. Anti-CD38 monoclonal antibodies (e.g., daratumumab, isatuximab) and SLAMF7-targeted antibodies (elotuzumab) are highly effective in inducing deep remissions and achieving minimal residual disease (MRD) negativity, even in patients with renal impairment [24]. The possibility of administration in patients with impaired kidney function makes them particularly relevant in patients who may become eligible for kidney transplantation.

**Cellular therapies:** The most recent advances include chimeric antigen receptor T-cell (CAR-T) therapy and bispecific T-cell engagers, both showing unprecedented efficacy in heavily pretreated MM. CAR-T therapies directed against B-cell maturation antigen (BCMA) achieved high rates of stringent complete response and durable MRD negativity [25]. Although data on their use in patients with advanced kidney disease are still limited, their potential to induce long-lasting remission raises the possibility of expanding transplant eligibility to previously excluded patients.

### 4.2. Hematopoietic Stem Cell Transplantation

HSCT plays an important role in consolidating hematological response and achieving long-term disease control. In fit patients eligible for HSCT, autologous HSCT continues to be part of the standard of care [26]. Although both AKI and acute kidney disease are known complications that may occur during this procedure [27,28], patients with MM-related kidney disease can safely undergo autologous HSCT. Considering that sustained hematological remission with MRD negativity reduces the risk of relapse under post-transplant immunosuppression, this autologous HSCT is particularly relevant in the context of kidney transplantation.

Allogeneic HSCT is associated with significant treatment-related morbidity (graft-versus-host disease, opportunistic infections, secondary malignancies), which makes it reserved for highly selected patients, such as those with high-risk cytogenetics, aggressive relapsed disease, or early relapse after autologous HSCT. Given its risks, allogeneic HSCT is rarely considered in patients who may also require kidney transplantation, although sequential or combined transplant strategies have been explored in small series.

### 4.3. Implications for Kidney Transplant Eligibility

This revolution in MM treatment and consequent impact on prognosis has turned MM into a manageable chronic disease, allowing many patients to achieve durable remission.

Considering the need for immunosuppression following kidney transplant and its relationship with higher risk of relapse, achieving complete remission or very good partial response (VGPR) along with MRD negativity after autologous HSCT is an important determinant of transplant eligibility [11,12].

In this new era of MM treatment, many options (e.g., proteasome inhibitors, IMiDs, and monoclonal antibodies) not only induce durable responses but are also safe for kidney transplant recipients [8]. At the same time, the revolutionary results for high-risk disease or relapsed/refractory MM brought by cellular immunotherapies, such as CAR-T and bispecific antibodies, have raised the hope of eventually offering kidney transplantation to even more patients in the future.

## 5. Kidney Transplantation in Patients with Multiple Myeloma

### 5.1. Rationale for Kidney Transplantation in Patients with Multiple Myeloma

In 2022, Jia H. Ng et al. concluded that kidney transplant recipients with plasma cell dyscrasias-related ESRD had worse overall and graft survival when compared with patients without plasma cell dyscrasias. On the other hand, as previously mentioned, recent advancements have led to a significant improvement in the overall survival of patients with MM [29]. The incidence and prevalence of patients with MM and ESRD is increasing [30]. Long-term dialysis in these patients not only may limit their access to new treatment options and clinical trials but is also associated with worse quality of life and survival [31].

Furthermore, the growing number of individuals on the kidney transplant waiting list and scarcity of organs warrant a careful and multidisciplinary approach to select the patients that will benefit the most from the procedure. Therefore, it is crucial that the scientific community consider strategies, protocols, and updated guidelines that facilitate the selection of these MM patients.

### 5.2. Historical Case Reports

In the pre-proteasome inhibitors era, evidence is scarce and limited to case reports and case series with generally poor outcomes regarding both overall and disease-free survival.

In 1997, Penn et al. published retrospective data from the Israel Penn International Transplant Tumor Registry considering kidney transplant outcomes in patients with previous neoplasms, among whom 12 patients had MM. Eight of these 12 patients (67%) relapsed post-transplant [32]. Leung et al. (2004) evaluated long-term outcomes of 7 patients with Light Chain Deposition Disease submitted to kidney transplantation; 5 patients had disease relapse identified by graft biopsy, and 4 patients died [33]. Tsakiris et al. (2010) analyzed data from a European ESRD registry from 1986 to 2005 and reported 35 patients with previous plasma cell dyscrasia that were submitted to kidney transplantation with an unadjusted median survival of 9.6 years, contrasting with the 19.6 years of the non-MM patients [30]. Nevertheless, these reports of disease relapse may be associated with outdated MM treatment strategies in patients lacking cytogenetic staging and without proper hematological disease control. Therefore, a causal association between disease relapse and kidney transplantation cannot be assumed.

### 5.3. Current Evidence

With the appearance of newer drugs and treatment strategies, overall MM outcomes improved, transforming what was once a potentially deadly disease into a chronic disease. In this new era, several groups began submitting patients with MM with sustained remission/controlled disease and ESRD to kidney transplantation with reasonable outcomes.

Huskey et al. (2018) [8] published their experience with 4 kidney transplant recipients with MM. One of the patients died with progressive hematological disease 5.5 years post-transplant; one patient presented with acute allograft rejection 10 months after kidney transplantation; and one patient had disease recurrence 8 months after kidney transplantation and was successfully treated. However, all 4 of these patients were submitted to different MM treatment strategies, obtained different disease responses, and underwent different immunosuppression schemes.

Korman et al. (2019) [34] compared 13 patients with MM and ESRD submitted to kidney transplantation with 65 matched controls and found similar graft (median 80 months) and patient survival (median 117 months), as well as a significant improvement in overall survival when compared with control hemodialyzed patients. They also reported higher viral and fungal infection rates, higher rates of hypogammaglobulinemia, and two cases of graft loss. These results suggest a combined effect of immunosenescence, adverse effects from hematological treatment, and specific immune defects regarding viral and fungal resistance in both the innate and the adaptive immune response in MM patients [32]. Although this higher risk of infection was not consistent in all the studies, it is safe to assume these patients warrant tailored management of immunosuppressors and antimicrobial prophylaxis.

In 2022, Mayo Clinic published their experience of 12 kidney transplants in MM patients between 1994 and 2019. Upon transplantation, eight had a complete hematological response, two had VGPR and two had partial responses. Seventy-five percent of patients had hematological disease progression, 25% lost allograft function, and approximately 46% died with a functional graft. Five-year allograft survival was 66%, progression-free survival was 44%, and overall survival was 61%. It is also worth mentioning that overall survival of patients submitted to a bortezomib-based treatment was statistically significantly higher at one, three, and five years when compared to those with bortezomib-free schemes [31].

### 5.4. Kidney Transplantation in Particular Situations

Sayed et al. (2015) [35] examined data from the National Amyloidosis Center, which included seven cases of monoclonal immunoglobulin deposition disease submitted to kidney transplantation. Three patients lost their allograft—two due to disease recurrence and the other due to allograft rejection. Patients that evolved with disease recurrence had not been submitted to plasma cell dyscrasia therapy. Kourelis et al. (2016) [36] published a cohort of nine patients with monoclonal immunoglobulin deposition disease submitted to kidney transplantation between 1992 and 2014. Three patients had disease recurrence on the renal graft, among whom, one patient had not been submitted to plasma cell dyscrasia therapy, and one had not achieved hematological response prior to the kidney transplant.

Molina-Andújar et al. (2021) also analyzed a cohort of patients with previously treated monoclonal immunoglobulin deposition disease submitted to kidney transplantation between 2010 and 2019 [37]. They included six patients previously treated with bortezomib, melphalan, and autologous HSCT who achieved complete hematological response. After kidney transplantation, one patient had a hematological relapse, and two patients presented disease progression [37].

Havasi et al. (2022), in collaboration with the International Kidney and Monoclonal Gammopathy Research Group, published a multicentric study including 237 patients with AL-amyloidosis submitted to kidney transplantation between 1987 and 2020 [38]. Median overall survival was 8.6 years and median graft survival 7.8 years, with significantly longer survivals in patients that had achieved complete response and VGPR. Recurrence was associated with hematological response. In the disease relapse subgroup, 87% of patients were successfully treated, preventing renal allograft loss [38].

Similarly to what was described regarding MM, available evidence supports kidney transplantation in carefully selected patients with monoclonal immunoglobulin deposition disease or AL amyloidosis, and efforts should be made to achieve complete hematological response or a VGPR.

### 5.5. Combined Approaches

The most consensual combined strategy includes chemotherapy followed by autologous HSCT to achieve complete MM remission before submitting the patient to kidney transplantation. Recently, a review from Sethi et al. (2025) focused on studies from the “Bortezomib era” of MM patients submitted to kidney transplant after having had an autologous HSCT [39]. From a total of 18 patients, five patients presented with VGPR and nine with complete response. Five patients relapsed, three had allograft rejections, and three lost their kidney graft. Overall survival rate at five years was 50% [39].

Other strategies involve submitting the MM patient to a combined allogeneic HSCT and kidney transplant from the same donor. The rationale for this therapeutic scheme is intentionally inducing a chimerism at the moment of the transplant, therefore potentially reducing immunological risk [8]. Baraldi et al. (2016) reported a series of 13 patients [40]. From these, one patient died from hematological recurrence, three achieved partial remission, and nine achieved a complete disease response. Also noteworthy is the fact that immunotolerance was achieved in 10 patients, allowing for the complete withdrawal of immunosuppressors, even though 6 patients had to resume it because of graft-versus-host disease [40].

Both the abovementioned strategies seem promising; nonetheless, the combined HSCT and kidney transplant from the same donor, given the scarcity of potential donors, does not seem feasible in current clinical practice.

The case series and cohorts retrieved during this review are summarized in Table 1 and Table 2.

## 6. Eligibility Criteria for Kidney Transplantation in Multiple Myeloma

### 6.1. Hematological Criteria

**Cytogenetic profiling.** High-risk cytogenetic abnormalities, commonly referred to as high-risk cytogenetics, are the most widely recognized prognostic indicators of poor outcomes in MM. The detection by FISH of t(4;14), t(14;16), del(17p), 1q gain [gain(1q)], and 1p deletion [del(1p)] has been consistently associated with both lower disease-free survival and overall survival despite intensive therapy [42]. Thus, inclusion of patients with high-risk cytogenetics in kidney transplant programs is still controversial, and units that may include them demand MRD negativity.

Assessing **hematological response** after treatment is crucial in order to define the risk of relapse over time and, consequently, to weigh kidney transplant benefits for these patients. The International Myeloma Working Group has presented as hematological response the following:

A complete response (CR)—negative immunofixation on both serum and urine, disappearance of any soft tissue plasmacytomas, and <5% plasma cells in the bone marrow aspirate [43]. Although CR reflects profound disease control, it does not exclude the persistence of clonal plasma cells below the detection limits of standard assays.

A very good partial response (VGPR)—residual M-protein detectable only by immunofixation but not electrophoresis, or a ≥90% reduction in serum M-protein with urine M-protein < 100 mg/24 h [43]. VGPR represents a deep remission and has often been used as a threshold for considering inclusion in a kidney transplant program in standard-risk patients.

Minimal residual disease (MRD) negativity—absence of clonal plasma cells on next-generation flow cytometry or sequencing with sensitivity ≥ 10^−5^ [44]. It is now considered the most sensitive marker of long-term prognosis and represents the optimal hematological status for kidney transplant eligibility.

Finally, **functional status** must be adequate to tolerate transplantation. This is typically assessed by the Karnofsky Performance Status (KPS ≥ 70%) or Eastern Cooperative Oncology Group (ECOG) Performance Status score (0–2) [45]. Performance status provides a global measure of resilience and treatment tolerability.

### 6.2. Nephrological Criteria

The 2020 KDIGO kidney transplant guidelines consider for the first time the inclusion of MM patients in kidney transplant programs, provided that stable remission following potentially curative therapy has been achieved [2,46].

Apart from hematological remission, patients are required to be in a fit condition, as for any other candidate. Patients must demonstrate the following:Clinical stability and absence of active infection, absence of active neoplasms;Absence of uncontrolled comorbidities;Capacity for adherence to immunosuppressive therapy and follow-up [2].

### 6.3. Timing for Kidney Transplantation

The optimal timing for undergoing kidney transplantation in MM is yet to be defined. Prolonged waiting periods reduce the number of quality-of-life years provided by the kidney transplant, whereas proceeding too early might unmask occult relapse and compromise graft outcomes [47].

A waiting period of **6–12 months** after HSCT and remission achievement is often recommended. This interval traduces stratification by risk category, as follows:**Standard-risk, MRD-negative patients** may be considered after 6 months;**Standard-risk, MRD-positive patients or high-risk, MRD-negative patients** are advised to wait at least 12 months;**High-risk, MRD-positive patients without remission** are unlikely to benefit from kidney transplant.

This approach aims to balance timely transplantation with adequate confirmation of durable disease control.

### 6.4. Multidisciplinary Context

The ultimate aim of kidney transplant is to improve survival and quality of life in ESRD patients [48]. MM patients should only be considered for this renal replacement therapy if there is a reasonable expectation of long-term survival [8]. For this decision, it is necessary to take into consideration other comorbidities that can have higher incidence in MM patients, such as cardiovascular disease [49,50], comorbid diabetes [51], and frailty [52], while prior or secondary malignancies may complicate management [53].

Other non-medical requirements for a successful kidney transplant are economic and psychosocial stability (instability is linked to nonadherence, infection, rejection, and graft loss, whereas strong social support networks improve adherence, reduce complications, and enhance quality of life) [54].

In summary, eligibility for kidney transplantation in MM must be defined through a multidisciplinary evaluation integrating hematological response, conventional kidney transplant criteria, timing of transplantation, comorbidities, psychosocial support, and patient preferences. By applying structured and individualized criteria, clinicians can identify those patients most likely to benefit from kidney transplantation (Table 3).

## 7. Proposed Algorithm for Patient Selection and Follow Up

We propose a stepwise algorithm that emphasizes disease control, clinical stability, optimal timing for kidney transplantation, and post-transplant follow-up (Figure 1).


**Step 1—Hematological assessment.**


Confirm remission status. Consider only patients with ≥VGPR; MRD negativity and standard-risk cytogenetics are preferred. Document the depth and date of the last hematologic response and exclude any evidence of active disease.


**Step 2—Nephrology assessment and multidimensional eligibility check.**


Evaluate performance status (ECOG 0–2 or Karnofsky ≥ 70%), optimize comorbidities (e.g., cardiovascular disease and diabetes), and confirm the absence of active infection or malignancy. Ensure immunizations are up to date and that transplant-specific risks (e.g., vascular access issues and urinary tract abnormalities) are addressed. Conduct a psychosocial assessment: adherence history, social support, psychiatric stability, and the patient’s willingness to engage in shared decision-making.


**Step 3—Post-treatment interval (timing).**


Ensure an adequate waiting period after hematologic therapy/HSCT:Standard-risk, MRD-negative: consider ≥6 months post-therapy/HSCT.High-risk cytogenetics or MRD-positive: wait ≥12 months.

Any biochemical/clinical relapse restarts the clock.


**Step 4—Kidney transplant.**


Proceed if all prior steps are met. Individualize the immunosuppression and infection-prophylaxis plan in coordination with hematology, with explicit documentation of a strategy for relapse management if needed.


**Step 5—Post-transplant monitoring.**


Quarterly serum protein electrophoresis/immunofixation and serum free light chains (FLCs); routine renal function and proteinuria surveillance; bone marrow reassessment when biochemical or clinical triggers arise. Maintain guideline-based infection and secondary malignancy screening. If relapse occurs, promptly re-evaluate immunosuppression and initiate appropriate anti-myeloma therapy.

## 8. Immunosuppression and Maintenance Strategies

### 8.1. Challenges of Immunosuppression in Multiple Myeloma

The management of immunosuppressive therapy in MM patients, both before and after kidney transplantation, requires thorough consideration. Two major concerns include the increased risk of infections and the potential for hematological relapse [55].

**Increased Risk of Infections**. Patients with MM are already more predisposed to infections due to humoral immunoparesis, associated with hypogammaglobulinemia and dysfunction of B cells, CD4+ T cells, and natural killer cells. The infectious risk is further enhanced by the myelosuppressive effects and viral reactivation associated with anti–plasma cell therapies [56,57]. This higher risk of infection is more pronounced in some subgroups, such as the elderly, patients with poor performance status, and individuals with significant comorbidities such as diabetes mellitus [57]. Moreover, combination regimens (e.g., proteasome inhibitors with IMiDs agents or monoclonal antibodies) significantly increase the incidence of severe infections, with reported odds ratios between 10 and 17, particularly after multiple lines of treatment [58].

**Potential Risk of Relapse**. There is a gap in knowledge regarding the relationship between immunosuppressive therapy and hematologic relapse. It is estimated that the immunologically dysregulated microenvironment promotes malignant plasma cell survival and clonal expansion by favoring cytokines such as IL-6, IL-10, TGF-β, and VEGF [59]. Long-term use of corticosteroids and other immunosuppressants may further impair immune surveillance against MRD, playing an important role in disease recurrence [59].

### 8.2. Strategies for Maintenance Therapy

The best strategy for maintenance immunosuppression and anti-MM therapy after kidney transplantation remains unclear. In these patients, it is essential to maintain hematological disease control while ensuring graft protection and avoiding excessive immunosuppression. In low-risk MM patients who achieve sustained MRD negativity after kidney transplantation, omission of maintenance MM therapy may represent a reasonable and potentially safe strategy [39].

**Immunomodulatory Drugs.** IMiDs such as lenalidomide have unquestionably improved survival in MM, but their use in solid organ transplant recipients is problematic. Lenalidomide acts through enhancement of natural killer cell cytotoxicity, increases CD3+/CD8+ T cells, and augments interferon-γ secretion, while inhibiting the PD-1/PD-L1 axis. These effects are directly linked to acute inflammation and graft rejection [8].

**Proteasome Inhibitors.** Proteasome inhibitors have shown promising results as an alternative maintenance therapy [8]. The use after autologous HSCT has been associated with improved progression-free survival and has not been associated with an increased risk of secondary malignancies compared with placebo or thalidomide [60]. Additional studies support superiority in terms of efficacy of proteasome inhibitors as maintenance therapy when compared with IMiDs, demonstrating better response rates, progression-free survival, and overall survival [61].

**Targeted Therapy in t(11;14).** The t(11;14) cytogenetic abnormality is associated with BCL-2 dependence, making venetoclax a potentially effective therapeutic option in this subgroup of patients [62]. Venetoclax, an oral selective BCL-2 inhibitor, has demonstrated high overall response rates (up to 92%) in cases of relapsed/refractory disease, with a median progression-free survival of 10 months and overall survival of 14.6 months [8,62]. While current results are promising, its role in post-transplant maintenance therapy still requires further investigation.

### 8.3. Proposed Monitoring Strategies

**Surveillance of Disease Markers.** After kidney transplant, patients with MM should be carefully monitored. Serum protein electrophoresis and FLCs quantification should be assessed at least every three months, with increased frequency in case of suspected relapse [8,63].

**Infection Prophylaxis and Screening.** Infection prophylaxis according to available guidelines is essential and should include vaccination against encapsulated bacteria and respiratory viruses, as well as prophylactic antibiotic, antiviral, and antifungal strategies. Opportunistic infections and secondary malignancies screening is mandatory [64]. A multidisciplinary approach with close collaboration between the nephrology and hematology departments is recommended [8].

**Adjustment of Immunosuppression in Relapse.** In a situation of hematological relapse, continuation of intensive immunosuppression may delay the efficacy of salvage therapy and increase toxicity risks [65]. In this scenario, previous therapies, treatment-related toxicities, and prognostic factors should be re-evaluated. Immunosuppression should be modified, including adjustment of corticosteroid dose or even immunosuppressive treatment discontinuation. Simultaneously, rescue or maintenance anti-MM therapy should be initiated in order to achieve disease control [66] (Figure 2).

## 9. Discussion and Conclusions

The option of kidney transplant for patients with MM is a recent concept that results from deeper knowledge of cytogenetics, emerging treatment options, and a progressively better understanding of transplant immunology, and available evidence, although largely based on small retrospective series and registry analyses, suggests that transplantation can be feasible and beneficial in carefully selected patients.

Optimal candidates are those with standard-risk cytogenetics, sustained remission after modern therapy (including autologous HSCT), and adequate performance status. These studies consistently point to hematological response depth post-transplant outcomes, highlighting the role of MRD assessment.

Despite encouraging reports, most studies are retrospective, single-center, and involve heterogeneous populations with variable MM therapies and follow-up durations. These limitations preclude strong recommendations on the timing of transplantation, optimal waiting periods after HSCT, or the best immunosuppressive regimens. The impact of maintenance MM therapy on graft outcomes is not known. KDIGO 2020 guidelines acknowledge transplantation as an option for MM patients in remission but do not specify thresholds for remission depth or timing. Our synthesis suggests that integration of cytogenetic risk profiling and MRD assessment into candidate evaluation is essential for clinical decision-making.

Future priorities must include prospective multicenter studies and international registry initiatives to address the above-mentioned challenges. Research on biomarkers, such as circulating tumor DNA and immune profiling, may further improve risk prediction and post-transplant monitoring. Ultimately, a multidisciplinary approach integrating hematology, nephrology, and transplant expertise will be essential to ensure that kidney transplantation benefits this high-risk but increasingly treatable population.

## Figures and Tables

**Figure 1 ijms-26-09358-f001:**
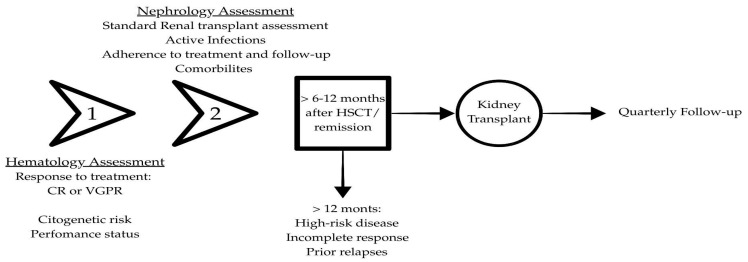
Stepwise algorithm integrating hematologic response, comprehensive transplant eligibility, optimal timing after therapy, and structured post-transplant surveillance.

**Figure 2 ijms-26-09358-f002:**
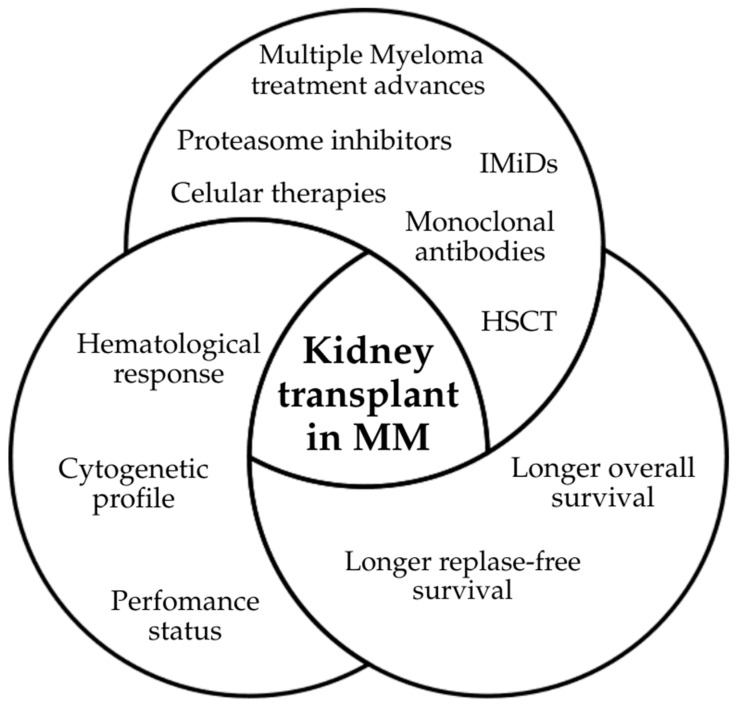
Main advances that contributed to considering kidney transplant possibility in MM patients. Legend: This figure summarizes the therapeutic and diagnostic breakthroughs that have expanded eligibility for kidney transplantation in MM: Proteasome inhibitors (e.g., bortezomib, carfilzomib, and ixazomib): Effective even in advanced kidney disease, improving renal recovery and overall survival. Immunomodulatory drugs (thalidomide, lenalidomide, and pomalidomide): Extended patient survival, though careful use is needed in transplant settings due to rejection risk. Monoclonal antibodies (anti-CD38 and SLAMF7): Induce deep hematologic responses and can be used safely in patients with renal impairment. Cellular therapies (CAR-T, bispecific T-cell engagers): Provide long-lasting remission in heavily pretreated patients, potentially broadening transplant eligibility. Autologous HSCT consolidation: Achieving MRD negativity post-HSCT reduces relapse risk under immunosuppression. Cytogenetic risk stratification (FISH-based profiling): Identifies high-risk vs. standard-risk patients, guiding transplant decision-making. Improved infection prophylaxis and transplant immunology: Advances in supportive care have reduced infectious complications and improved graft outcomes. Together, these innovations have transformed MM into a more chronic and manageable condition, allowing kidney transplantation to be considered in selected patients who achieve durable remission.

**Table 1 ijms-26-09358-t001:** Kidney transplantation in multiple myeloma.

Reference	Number of Patients	Renal Lesion	Pre-Transplant Treatment (Chemo/Autologous HSCT)	Main Outcomes
Humphrey 1975 (Pre-bortezomib) [41]	1	LCCN	Chemotherapy	Patient died from infection at 3 months; graft outcome not specified
Penn 1997 (Pre-bortezomib) [32]	12	MM (various)	Various	8/12 relapsed (67%); poor overall patient survival (most died within a few years); graft survival variably reported
Leung 2004 (Pre-bortezomib) [33]	7	LCDD	Melphalan + prednisone (3 patients)	Median graft survival 11 months; 5/7 relapsed; 4 patients died; 1 patient alive with functioning graft at 13 years
Tsakiris 2010 (Pre-bortezomib) [30]	35 (from 2453 RRT)	MM/LDD	Not specified (1986–2005)	Mean overall patient survival 9.6 years (vs 19.6 years in non-MM); graft survival not reported
Huskey 2018 (Modern era) [8]	4	MM	Bortezomib, Lenalidomide, thalidomide, dexamethasone; 2 autologous HSCT	Follow-up 46 months; 1 death at 5.5 years (patient survival); 1 acute rejection at 10 months (graft loss); 1 hematological relapse at 8 months without renal relapse; 1 stable
Kormann 2019 (Modern era) [34]	13 (vs. 65 controls)	MM/SMM, LCDD, AL amyloidosis	Chemotherapy ± autologous HSCT	Median patient survival 117 months; median graft survival 80 months; no significant difference vs. controls; higher infection rates
Heybeli 2022 (Modern era) [31]	11 (12 kidney transplants)	MM	Bortezomib-based ± lenalidomide; 8 CR, 2 VGPR, 2 PR	5-year graft survival 66%; 5-year progression-free survival 44%; 5-year overall patient survival 61%; 75% hematological relapse; 25% graft failure
Ng 2022 (Modern era) [29]	National United States cohort	MM and amyloidosis	Registry data	Overall patient survival worse in plasma cell dyscrasias vs. non-PCD; outcomes worse in amyloidosis; MM survival closer to controls
Sethi 2025 (Modern era) [39]	18	MM post-Autologous HSCT	Autologous HSCT + bortezomib-based chemotherapy	Median interval HSCT to transplant: 29.5 mo; pre-transplant 5 VGPR, 6 CR, 5 sCR; relapse rate 27.7%; rejection 16.6%; graft loss 16.6%; 5-year overall patient survival 50%

Legend: CR = complete remission; VGPR = very good partial remission; PR = partial remission; sCR = strict complete remission; HSCT = hematological stem cell transplant; LCCN = light chain cast nephropathy; LCDD = light chain deposition disease; PCD = plasma cell dyscrasia; RRT = renal replacement therapy.

**Table 2 ijms-26-09358-t002:** Kidney transplantation in monoclonal immunoglobulin deposition disease.

Reference	Number of Patients	Renal Lesion	Pre-Transplant Treatment (Chemo/Autologous HSCT)	Main Outcomes
Leung 2004 (Pre-bortezomib) [33]	7	LCDD	Melphalan + prednisone (3 patients)	Median follow up 33.3 months; 5/7 relapse; mean graft loss 11 months; 4 deaths; 1 alive 13 years
Sayed 2015 (Modern era) [35]	7	LCDD	4 chemotherapy/Autologous HSCT; 3 none	3 graft losses (2 relapse at 1.6 and 1.9 yrs, 1 rejection); 4 functioning grafts (eGFR > 40 mL/min)
Kourelis 2016 (Modern era) [36]	9	LCDD	Chemotherapy ± bortezomib	3 graft recurrences (2–9 yrs); 1 bortezomib + CR relapsed at 9 years; others without hematological response relapsed earlier
Molina-Andújar 2021 (Modern era) [37]	6	LCDD	Bortezomib + melphalan + Autologous HSCT	All CR pre-transplant; follow up 20.5 months; 1 hematological relapse; 2 disease progressions (1 graft loss); 5 patients alive with functioning grafts

Legend: CR = complete remission; HSCT = hematopoietic stem cell transplant; eGFR = estimated glomerular filtration rate; LCDD = light-chain deposition disease.

**Table 3 ijms-26-09358-t003:** Simplified approach to kidney transplantation in multiple myeloma.

Category	Green (Eligible/Favorable)	Yellow (Conditional/Caution)	Red (Contraindicated/Unfavorable)
Hematological Response	Complete response (CR) or very good partial response (VGPR) with MRD negativity; standard-risk cytogenetics	VGPR with MRD positivity high-risk cytogenetics but in sustained remission	Active disease high-risk with persistent MRD positivity
Timing	≥6 months post-HSCT with stable remission (standard risk, MRD-negative)	12 months post-HSCT for high-risk or MRD-positive patients	<6 months from therapy Relapsed/refractory disease
Performance Status	ECOG 0–2 or KPS ≥ 70%	ECOG 2–3 with potentially reversible frailty/comorbidities	ECOG ≥ 3 Severe, irreversible frailty
Comorbidities	Controlled cardiovascular disease Well-managed diabetes No active infections or cancer	Controlled but significant comorbidities (e.g., stable ischemic heart disease, mild frailty)	Severe cardiovascular disease Uncontrolled diabetes Active infection/other cancers
Psychosocial factors	Strong social support Demonstrated adherence to therapy and follow-up	Limited support systems but improving with intervention	Active substance abuse Major untreated psychiatric illness
Patient preferences	Fully informed and motivated; engaged in shared decision-making	Uncertain adherence or ambivalence	Refusal of therapy Inability to comply with follow-up

CR = complete response; VGPR = very good partial response; HSCT = hematopoietic stem cell transplant; ECOG = Eastern Cooperative Oncology Group Performance Status; KPS = Karnofsky performance status; MRD = minimal residual disease. Legend: Green = favorable conditions where kidney transplantation is recommended. Yellow = intermediate situations requiring individualized, cautious decision-making. Red = unfavorable conditions where transplantation is contraindicated. Sources: Adapted from KDIGO 2020 guidelines [2,46], Sethi et al., 2025 [39], and synthesis of available cohort and registry evidence.

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
