# Peer review of "Kidney Transplantation in Patients with Multiple Myeloma: Current Evidence, Challenges, and Future Directions"

_ijms, 2025, doi:10.3390/ijms26199358_

Round 1

Reviewer 1 Report

Comments and Suggestions for Authors

This manuscript, “Kidney Transplantation in Patients with Multiple Myeloma: Current Evidence, Challenges and Future Directions,” addresses a timely and clinically significant topic at the intersection of nephrology and hematology. The authors provide a comprehensive narrative review of recent therapeutic advances in multiple myeloma and their impact on kidney transplant eligibility and outcomes. The subject is of considerable importance because improved myeloma therapies have expanded the pool of potential kidney transplant candidates, yet guidance on patient selection, timing, and post-transplant management remains limited. While the review is well organized and draws on a broad literature base, several key areas require clarification and refinement to enhance the manuscript’s scientific rigor and clinical applicability.

  1. The abstract is concise and well written, but it reads more like a background section than a structured summary. Please add a one-sentence statement of the main conclusion
  2. Consider including key quantitative outcomes in abstract (graft survival, relapse rates) to strengthen the abstract’s impact.
  3. Introduction is comprehensive but somewhat repetitive (lines 34–46 and 47–52 cover similar rationale). A tighter paragraph structure would improve readability.
  4. Add a clear statement of review methodology (databases searched, years covered) to introduction to justify the literature selection.
  5. As a narrative review, the manuscript lacks an explicit “Methods” section. Please describe your search strategy: databases (PubMed, Embase), key terms, inclusion/exclusion criteria, and date of last search. This is important for reproducibility and transparency.
  6. Include follow-up duration medians and ranges where available.
  7. Clarify if survival percentages are graft survival or patient survival to avoid confusion.
  8. Some statements could use more precise referencing (e.g., “75% of patients had hematological disease progression” – specify study cohort size again for context).
  9. In discussion there is a good overview of clinical implications, but several paragraphs reiterate points from the Results. Consider merging or condensing for conciseness.
  10. Add a short comparison with existing international guidelines (KDIGO 2020) highlighting points of agreement or divergence.
  11. To increase impact, I suggest adding a forward-looking statement to conclusion on required prospective studies or registry efforts.
  12. These changes will substantially improve the manuscript’s clarity, reproducibility, and value for the IJMS readership.

Author Response

Response to Reviewers – IJMS – 3877397

Title: Kidney Transplantation in Patients with Multiple Myeloma: Current Evidence, Challenges and Future Directions

We thank the reviewers and editors for their thoughtful and constructive feedback. We carefully revised the manuscript in response to all comments, which has improved its clarity, rigor, and clinical relevance. Below we provide a point-by-point response.

Reviewer comments are in italics, followed by our replies.

Response to Reviewer 1

This manuscript, “Kidney Transplantation in Patients with Multiple Myeloma: Current Evidence, Challenges and Future Directions,” addresses a timely and clinically significant topic at the intersection of nephrology and hematology. The authors provide a comprehensive narrative review of recent therapeutic advances in multiple myeloma and their impact on kidney transplant eligibility and outcomes. The subject is of considerable importance because improved myeloma therapies have expanded the pool of potential kidney transplant candidates, yet guidance on patient selection, timing, and post-transplant management remains limited. While the review is well organized and draws on a broad literature base, several key areas require clarification and refinement to enhance the manuscript’s scientific rigor and clinical applicability.

Response: We thank the reviewer for the positive feedback.

Comments 1.  The abstract is concise and well written, but it reads more like a background section than a structured summary. Please add a one-sentence statement of the main conclusion 

Response 1. We thank the reviewer and added a concluding sentence to the abstract (- In conclusion, kidney transplantation may be a feasible option for carefully selected MM patients achieving deep and sustained remission, though prospective data are still needed).

Comments 2.  Consider including key quantitative outcomes in abstract (graft survival, relapse rates) to strengthen the abstract’s impact.

Response 2. We revised the Abstract to include key outcomes from the largest series (Reported 5-year allograft survival ranges from 50-66, progression-free survival of 44%, and overall survival of 61%, depending on pre-transplant remission depth).

Comments 3.  Introduction is comprehensive but somewhat repetitive (lines 34–46 and 47–52 cover similar rationale). A tighter paragraph structure would improve readability.

Response 3. We reorganized all the introduction section to avoid repetition, by merging the two paragraphs referred above (updated text in the manuscript).

Comments 4.  Add a clear statement of review methodology (databases searched, years covered) to introduction to justify the literature selection. Comments 5. As a narrative review, the manuscript lacks an explicit “Methods” section. Please describe your search strategy: databases (PubMed, Embase), key terms, inclusion/exclusion criteria, and date of last search. This is important for reproducibility and transparency.

Response 4 and 5. We agree this comment related to important information that was missing in our manuscript, so we decided to add a “Methods” section following introduction section. (Updated text in the manuscript - We performed a narrative review of the literature focusing on ESRD and kidney transplantation in patients with MM and related plasma cell dyscrasias. A systematic search was conducted in PubMed and Embase, identifying publications in English, Portuguese or Spanish, from January 1975 through June 2025. We used combinations of the keywords: “multiple myeloma,” “plasma cell dyscrasia,” “renal transplantation,” “kidney transplantation,” “end-stage renal disease,” and “monoclonal gammopathy”. We included case reports, case series, cohort studies, registry analyses, and review articles that provided clinical outcomes or guidance on patient selection, timing, or management of kidney transplantation in this population. Given the heterogeneity and limited number of available studies, we synthesized the evidence in a narrative format, supported by summary tables.)

Comments 6.  Include follow-up duration medians and ranges where available.

Response 6. We revised Tables 1 and 2 to include median follow-up times (and ranges where reported).

Comments 7.  Clarify if survival percentages are graft survival or patient survival to avoid confusion.

Response 7. Thank you for highlighting this aspect. We have clarified through the text (section 5.3) and we also included that information on table 1 and 2 in a more detailed way.

Comments 9.  In discussion there is a good overview of clinical implications, but several paragraphs reiterate points from the Results. Consider merging or condensing for conciseness.  Comments 10.   Add a short comparison with existing international guidelines (KDIGO 2020) highlighting points of agreement or divergence. Comments 11. To increase impact, I suggest adding a forward-looking statement to conclusion on required prospective studies or registry efforts.

Response 9-11. Thank you for your input on this section. After considering your comments and other reviewer´s comments on both discussion and conclusion sections we decided to merge discussion with conclusion in only one section to avoid repetition. This section now focuses on interpretation, limitations, comparison with guidelines, and future directions rather than restating results (updated text in the manuscript). We added this reference to KDIGO 2020 in the discussion.

Reviewer 2 Report

Comments and Suggestions for Authors

Dear Colleagues,

Your literature review is devoted to a highly relevant topic - the application of kidney transplantation in patients with multiple myeloma. You have analyzed a considerable number of contemporary references and provided your own synthesis. Moreover, you summarized the data in tables and figures, which is a valuable contribution to your work. At the same time, I have several comments regarding the manuscript.

Major comments.

1) References formatting and completeness.

The manuscript uses inconsistent referencing styles: in some places, both the author’s surname and year of publication are provided alongside the numerical reference (e.g., lines 124, 126, 131, 206, 222, 240, 247), while in others no references are given at all (e.g., lines 275–277, 281–283). Please unify the referencing style and ensure that all cited statements are properly referenced.

2) Table 3 — missing sources and unclear categorization.

Table 3 (line 390) does not contain references to the literature. Please clarify which sources were used for this summary. The meaning of the color coding in Table 3 is not explained. Does it relate to perception or classification? Please specify. The relationship between psychosocial factors and patient-related factors (lines 6–7) is not clearly justified. Please describe the objective criteria you applied in these assessments.

3) Patient selection algorithms.

The manuscript (line 294) mentions criteria for patient selection, but they are not clearly formulated. On what basis were these criteria established? Please provide a detailed explanation or a structured algorithm.

4) Figures - clarity and completeness.

Figure 1 (line 404) requires comments or explanations for stages 1, 2, 3, and 4. These should be clarified either in the text or in the figure legend. Figure 2 (page 14, line 442) is difficult to interpret: labels are too small, and the figure itself appears slightly blurred. Please improve the resolution and increase font size.

Minor comments.

1) Abbreviations.

The manuscript contains a large number of abbreviations, which complicates reading. I recommend including a list of abbreviations at the end of the manuscript. Even commonly used abbreviations (e.g., HSCT, line 175) should be spelled out upon first mention.

2) General formatting.

Please ensure overall uniformity in citation style and consistency in figure/table presentation throughout the manuscript.

Overall, I believe that addressing these comments will significantly improve the quality of the manuscript. The data summarized by the authors will be of considerable interest to specialists in clinical oncology, hematology, and nephrology. I wish you every success with your work.

Comments on the Quality of English Language

The English could be improved to more clearly express the research.

Author Response

Response to Reviewers – IJMS – 3877397

Title: Kidney Transplantation in Patients with Multiple Myeloma: Current Evidence, Challenges and Future Directions

We thank the reviewers and editors for their thoughtful and constructive feedback. We carefully revised the manuscript in response to all comments, which has improved its clarity, rigor, and clinical relevance. Below we provide a point-by-point response. Reviewer comments are in italics, followed by our replies.

Major comments.

Major comments 1.  References formatting and completeness. The manuscript uses inconsistent referencing styles: in some places, both the author’s surname and year of publication are provided alongside the numerical reference (e.g., lines 124, 126, 131, 206, 222, 240, 247), while in others no references are given at all (e.g., lines 275–277, 281–283). Please unify the referencing style and ensure that all cited statements are properly referenced.

Response 1. We have reviewed the document and uniformized the referencing style by providing the necessary references while maintaining the author’s name and year of publication.

Major comments 2. Table 3 — missing sources and unclear categorization. Table 3 (line 390) does not contain references to the literature. Please clarify which sources were used for this summary. The meaning of the color coding in Table 3 is not explained. Does it relate to perception or classification? Please specify. The relationship between psychosocial factors and patient-related factors (lines 6–7) is not clearly justified. Please describe the objective criteria you applied in these assessments.

Response 2. We revised Table 3, added explicit sources (KDIGO 2020, Sethi et al. 2025, and our synthesis), and included a clear legend. The meaning of color coding is now explained (Green = favorable, Yellow = caution, Red = contraindicated). Psychosocial factors were separated from patient preferences for clarity.

Major comments 3. Patient selection algorithms. The manuscript (line 294) mentions criteria for patient selection, but they are not clearly formulated. On what basis were these criteria established? Please provide a detailed explanation or a structured algorithm.

Major comments 4. Figures - clarity and completeness. Figure 1 (line 404) requires comments or explanations for stages 1, 2, 3, and 4. These should be clarified either in the text or in the figure legend. Figure 2 (page 14, line 442) is difficult to interpret: labels are too small, and the figure itself appears slightly blurred. Please improve the resolution and increase font size.

Response 3 and 4. We reorganized the section Proposed Algorithm and structured the text clarifying each step and giving detailed explanation. Concerning figure 2, we replaced it with a higher-resolution version; labels enlarged. Legend expanded to explain therapeutic advances and their relevance to transplantation.

Minor comments.

Minor comments 1. Abbreviations. The manuscript contains a large number of abbreviations, which complicates reading. I recommend including a list of abbreviations at the end of the manuscript. Even commonly used abbreviations (e.g., HSCT, line 175) should be spelled out upon first mention.

Response 1. Thank you for your suggestions. We added a list of abbreviations to the end of the manuscript. Regarding HSCT, it is first used and defined in the Introduction section, lines 39-40.

Minor comments 2. General formatting. Please ensure overall uniformity in citation style and consistency in figure/table presentation throughout the manuscript.

Response 2. We ensured consistent formatting across the manuscript: uniform reference style, figure quality, and table presentation.

Round 2

Reviewer 1 Report

Comments and Suggestions for Authors

I am satisfied with the authors’ responses to my previous comments and appreciate the revisions they have made to address the concerns raised. The manuscript is now significantly improved, and I believe it meets the standards for publication. I have no further concerns, and I recommend the paper for acceptance in its current form.